# Corneal Stroma Regeneration with Collagen-Based Hydrogel as an Artificial Stroma Equivalent: A Comprehensive In Vivo Study

**DOI:** 10.3390/polym14194017

**Published:** 2022-09-26

**Authors:** Egor Olegovich Osidak, Andrey Yurevich Andreev, Sergey Eduardovich Avetisov, Grigory Victorovich Voronin, Zoya Vasilievna Surnina, Anna Vladimirovna Zhuravleva, Timofei Evgenievich Grigoriev, Sergey Vladimirovich Krasheninnikov, Kirill Konstantinovich Sukhinich, Oleg Vadimovich Zayratyants, Sergey Petrovich Domogatsky

**Affiliations:** 1Imtek Ltd., 3rd Cherepkovskaya 15A, 121552 Moscow, Russia; 2Research Institute of Eye Disease, 11A Rossolimo St., 119021 Moscow, Russia; 3I.M. Sechenov First Moscow State Medical University, 8-2 Trubetskaya Str., 119991 Moscow, Russia; 4LEC Ltd., Rozhdestvenskaya, 7, 141006 Mytischi, Russia; 5FSBEI HE A.I. Yevdokimov MSMSU MOH Russia, Rakhmanovsky Lane, 3, 127994 Moscow, Russia; 6National Research Center Kurchatov Institute, Akademika Kurchatova pl.,1, 123182 Moscow, Russia; 7Koltzov Institute of Developmental Biology of the Russian Academy of Sciences, Vavilova 26, 119334 Moscow, Russia; 8FSBI National Medical Research Centre of Cardiology of the Ministry of Health of the Russian Federation, 3 Cherepkovskaya 15A, 121552 Moscow, Russia

**Keywords:** viscoll collagen membrane, cornea regeneration, stromal replacement, tissue engineering

## Abstract

Restoring the anatomical and functional characteristics of the cornea using various biomaterials is especially relevant in the context of a global shortage of donor tissue. Such biomaterials must be biocompatible, strong, and transparent. Here, we report a Viscoll collagen membrane with mechanical and optical properties suitable for replacing damaged stromal tissue. After removing a portion of the stroma, a Viscoll collagen membrane was implanted into the corneas of rabbits. After 6 months, the active migration of host cells into Viscoll collagen membranes was noted, with the preservation of corneal transparency in all experimental animals. Effective integration of the Viscoll collagen membrane with corneal tissue promoted nerve regeneration in vivo, as confirmed by in vivo confocal microscopy. We also demonstrated the safety and efficacy of the Viscoll collagen membrane for corneal stroma regeneration. Thus, in combination with the proposed packaging format that provides long-term storage of up to 10 months, this material has great potential for replacing and regenerating damaged stromal tissues.

## 1. Introduction

Throughout human life, the cornea is constantly in contact with its surrounding environment in a way that is harmful to the delicate tissue. The immediate environment includes high and low temperatures, air pollution and toxic substances, ultraviolet radiation, and other types of radiation. However, owing to its complex structure, which includes the unique nature of trophism, innervation, and a type of cellular regeneration, it remains transparent and is the main refractive lens of the eye [1].

Any damage or disease of the cornea can lead to a complete or partial loss of its properties; in particular, the loss of transparency leads to a significant decrease in visual acuity. The most common diseases include keratoconus, dystrophies of various origins, ulcers, and corneal injury [2].

Donor cornea transplantation is the most common method of surgical treatment for most diseases of the cornea; however, the global shortage of donor tissue significantly complicates its use. Thus, the development of alternative approaches based on tissue engineering and regenerative medicine is necessary to solve this problem [3].

Thus far, advances in the field of tissue engineering have shifted the therapeutic focus to creating an artificial cornea that produces the conditions for restoring specific layers through cellular regeneration [3]. This shift has become possible owing to the development of new, less invasive surgical techniques, including anterior and posterior lamellar keratoplasty, in which only the damaged area of the cornea is replaced, while leaving the surrounding healthy tissue intact [4]. These techniques can improve outcomes in terms of graft survival and the number of postoperative complications compared with penetrating keratoplasty, in which the entire cornea is replaced.

In our previous work, we demonstrated the safety and biocompatibility of a Viscoll collagen membrane based on medical-grade Viscoll collagen [5]. Implantation of this Viscoll collagen membrane resulted in an increased overall thickness of the cornea and its strength characteristics, as well as the maintenance of transparency for up to six months postoperatively. Importantly, the cornea is highly innervated, and nerve regeneration after injury plays a critical role in the restoration of normal function [6,7]. Therefore, it is essential to investigate the regeneration of the replaced corneal tissue and the innervation of the restored area.

The primary aim of this study is to conduct a comprehensive assessment of the suitability of the Viscoll collagen membrane as an artificial analogue of the corneal stroma for the regeneration of its defects. A feature of our manufacturing process of the collagen membrane was increasing the concentration of collagen instead of the standard chemical crosslinking protocols to improve the biomechanical characteristics of the final product. The specific objectives of the current work include the study of the biomechanical characteristics and transparency of the Viscoll collagen membrane in comparison with the stroma from the human cornea; toxicological studies of the Viscoll collagen membrane; and evaluation of the effectiveness of the Viscoll collagen membrane for corneal regeneration during its implantation in the cornea of rabbits, in which a section of the stroma was surgically removed.

## 2. Materials and Methods

### 2.1. Animal Experiments

All experiments were carried out in compliance with Directive 2010/63/EU and the Research Institute of Eye Diseases Animal Care and Use Committee guidelines, and the study was approved by the aforementioned institution’s review board (№763, date of approval 11 May 2021). The experimental studies involved 10 male Chinchilla rabbits, aged 12–16 weeks and weighing 3.0–3.5 kg. All animals were allowed to adapt to their environment for 2 weeks before surgery. The rabbits received general and local anesthesia before and during surgery. General anesthesia was administered via intramuscular injection of ketamine (50 mg/kg) and xylazine (15 mg/kg). After the surgery, the animals were maintained under controlled conditions: temperature, 22 ± 1 °C; relative humidity, 45%; 10 air changes per hour; and a 12 h light/dark cycle. The rabbits had free access to water and standard compound feed. The surgeries were performed under general anesthesia. The rabbits were euthanized under deep anesthesia (100 mg/kg ketamine and 4 mg/kg xylazine) with 80 mg/kg pentobarbital.

### 2.2. Collagen Membrane Preparation

The production of Viscoll collagen membrane was certified according to ISO 13485 (Quality System for Medical Devices). Viscoll Collagen Membrane was obtained sterile from initially sterile components as a result of an aseptic manufacturing process (according to ISO 13408) in ISO 5 and ISO 7 class cleanrooms (according to ISO 14644-1) and did not require final sterilization.

The Viscoll collagen membrane was prepared using a previously published method [5] consisting of two stages: gelation and vitrification at constant pressure.

#### 2.2.1. Gelation

Briefly, 0.5 mL of a sterile 30 mg/mL solution of medical grade native porcine collagen type I (Viscoll^®^; Imtek Ltd., Moscow, Russia) was added to each well of a 24-well plate. To remove air bubbles from the collagen solution, the plates were centrifuged for 30 min using an Avanti J-26XP centrifuge (oscillating bucket rotor JS-5.2; Beckman Coulter, Inc., Brea, CA, USA) at 3200 rpm and 4 °C. To induce collagen gelation, the plates were incubated in an atmosphere of ammonia vapor for 12 h at 20 °C. After incubation, collagen hydrogels were collected, immersed in water for injection (Solopharm, Saint-Petersburg, Russia), and incubated for 24 h at 20 °C.

#### 2.2.2. Plastic Compression and Vitrification

At this stage, collagen hydrogels were placed between two teflon plates, one of which was subjected to a constant load of 3 kg until complete vitrification.

#### 2.2.3. Packaging

For long-term storage, the resulting material was rehydrated in water for injection (Solopharm) for 5 min. Then, the collagen membrane was placed on a square Teflon plate and hermetically packed in sterile primary packaging that met the requirements of ISO 11607-1 (Clinipak^®^, Klinipak LLC, Moscow, Russia). The material in the primary package was hermetically packed under vacuum into a secondary vacuum package (Figure 1). The obtained material was stored at 4–10 °C for 10 months before use.

### 2.3. Human Corneal Stroma Samples Preparation

Sections of the human corneal stroma were obtained using a Moria Evolution 3E microkeratome (Moria, Antony, France). The human corneoscleral disc from healthy donors was placed in a chamber with connected irrigation and securely fixed. First, the surface layer of the cornea was removed, including the Bowman’s membrane, and then a stromal section of a given thickness was performed directly using a 150 μm microkeratome head. An ultrasonic pachymeter was used to measure the thickness of the central part of the cornea before and after the incision. As a result, stromal samples with an average thickness of 300 ± 20 µm were obtained.

### 2.4. Optical Properties of Collagen Membranes and Human Cornea

Light transmission through five Viscoll collagen membranes and five human corneal stroma samples were measured in the 380–780 nm wavelength range using a UV-VIS spectrophotometer (PE-5400UV, Ecroskhim, Saint-Petersburg, Russia). All samples were about 300 µm thick. Samples were immersed in 150 mM NaCl before measurement. During the measurement, samples were placed on one side of an empty transparent cuvette. An identical empty cuvette served as a reference.

### 2.5. Toxicology

Toxicological tests of the Viscoll collagen membrane were performed according to ISO 10993 standards by an independently certified testing laboratory center (IMBIIT, Moscow, Russia). Viscoll collagen membranes were tested according to the following protocols:ISO 10993-5: In vitro cytotoxicity. Direct effect of extracts from membranes on L929 mouse fibroblasts.ISO 10993-6: Implantation. The study is intended to evaluate the biocompatibility of the test material and tissue within the eye by surgical implantation of the material into the eye of a rabbit for an appropriate period of time. The reverse tolerance of the test material and eye tissues after implantation is evaluated.ISO 10993-10: Irritating effect. In vivo. Ocular irritation in rabbits.ISO 10993-10: Irritating effect. In vivo. Skin sensitization in guinea pigs.ISO 10993-11: Acute systemic toxicity test in rabbits.GPM.1.2.4.0005.15: Pyrogenicity. The test is based on the measurement of body temperature in rabbits before and after the injection of membrane extracts.

### 2.6. Ex Vivo Sewing Test

The cadaveric eye was fixed in a special holder. Non-penetrating trepanation of the cornea was performed at 2/3 of its depth, after which the upper layers of the stroma were removed, and a collagen graft was placed in the formed trepanation bed and fixed with a continuous 10-0 nylon suture. The detailed technique is presented in Appendix A.

### 2.7. Surgery

The rabbits received general and local anesthesia before and during surgery. General anesthesia was administered via intramuscular injection of ketamine (50 mg/kg) and xylazine (15 mg/kg). The local anesthesia (2% (*w*/*v*) lidocaine eye drops) was administered to the right eye of each rabbit. A mark was made on the cornea using an 8 mm-diameter trephine, and an approximately 3 mm-long incision was made along the marking line at 1/3 of the depth, with a disposable blade. An intrastromal pocket was formed through the formed access. To do this, a tunnel was formed along the notch line using a spatula, which was cut with scissors concentrically to the markup. Thus, intraoperative access was expanded by ½ of the circumference, and then, using a stratifier, we formed an intracorneal pocket. Repeated stratification was performed in the formed pocket, thereby highlighting the layer of the underlying stroma that was then removed. In one group of animals, a collagen membrane was implanted into the area where the stroma was excised (first group—surgery and implantation of membranes). In another group (second group—only surgery), consisting of two rabbits, implantation of the collagen membrane was not performed after the removal of the stroma, and the wound was sutured using 10-0 nylon sutures (MANI Inc., Utsunomiya, Japan). Eye drops containing 0.3% (*w*/*v*) gentamicin were applied daily for the first 3 days. The detailed technique is presented in Appendix A.

### 2.8. Postoperative Observation

Clinical evaluations of the corneas were performed on days 30, 90, and 180 post surgery using slit-lamp microscopy. On days 30, 90, and 180, anterior segment optical coherence tomography (OCT) and in vivo confocal microscopy (IVCM) were performed on the eight rabbits from the first group. The contralateral intact eyes served as the reference.

### 2.9. Histological and Immunohistological Study

The rabbits were sacrificed on the 180th day after surgery. Corneal samples were excised from four rabbits, fixed in 10% (*v*/*v*) neutral buffered formalin, dehydrated, and embedded in paraffin wax. Sections of dehydrated, paraffin-embedded samples (5 μm thick) were stained with hematoxylin and eosin using standard techniques and observed under an optical microscope.

For the immunohistochemistry analysis, fixed corneal samples from four rabbits in the first group and two rabbits in the second group and the contralateral intact eyes from both groups were washed in PBS and transferred to 30% (*w*/*v*) sucrose in PBS. Sections (14 μm thick) were obtained using a cryostat (CM1900, Leica, Weltzar, Germany). The sections were incubated for 1 h at 20–25 °C in blocking solution: 5% (*v*/*v*) normal goat serum (Sigma-Aldrich, St. Louis, MO, USA), 0.3% (*v*/*v*) Triton X-100, and 0.01 M PBS (pH 7.4). This was followed by incubation with mouse anti-alpha SMA (1:200; BioLegend, San Diego, CA, USA) in the blocking solution at 4 °C overnight. Thereafter, the sections were washed and incubated for 2 h with goat anti-mouse IgG (AlexaFluor488, 1:600; Abcam, Cambridge, UK) in 0.3% (*v*/*v*) Triton X-100 and 0.01 M PBS (pH 7.4). Sections were then washed in PBS, and nuclei were stained with DAPI solution (2 μg/mL; Sigma, St. Louis, MI, USA). Histological images were obtained using a BZ-9000E fluorescence microscope (Keyence, Osaka, Japan).

### 2.10. Mechanical Testing

The tensile strengths of the five collagen membranes were measured using an Instron Universal Testing Instrument (model 5982; Illinois Tool Works, Inc., Glenview, IL, USA) and a 2530-series load cell (model 2530-50N; Illinois Tool Works, Inc.). Mechanical trials were conducted in compliance with the ASTM D5748 standard. This testing method was designed to provide the conditions for biaxial deformation with a constant crosshead speed (1 mm/min). The specimen holder has a diameter of 5 mm. The loading element tip radius was 1.25 mm. During mechanical testing, all the samples were kept in Ringer’s solution at 20 °C. During the test, we considered the test samples as conical surfaces of variable height. The obtained experimental dependences in the load-displacement coordinates were recalculated into stress–strain curves using the following formulas.

Sample deformation:ε = ((S_tek_ − S_0_)/S_0_) × 100%,(1)
where S_0_ is the initial area of the sample and S_tek_ is the current area of the sample, calculated by the following formula: S_tek_ = πR (R^2^ + Δl^2^)^1/2^,(2)
where R is the sample radius and Δl is the displacement of the loading element.

The stress was calculated by the following formula:σ = P/S_tek_,(3)
where P is the current load on the sample.

Based on the stress–strain curves (Figure 2b), the fracture strain (at the maximum load) and the elastic modulus (in the section of the curve with the maximum slope) were calculated. All calculations were carried out using Instron Bluehill 2 Universal software (Illinois Tool Works, Inc.).

### 2.11. Statistical Analysis

The basic summary statistics are presented as the mean ± standard deviation. All statistical analyses were performed using Excel 16.4 (Microsoft, Inc., Redmond, DC, USA).

## 3. Results

### 3.1. Optical and Physical Properties

Figure 2a shows the mean of light transmission of five samples of membranes and five samples of human cornea stroma as a function of wavelength in the 380–780 nm range. The Viscoll collagen membrane can transmit visible light similar to the human cornea; however, the mechanical properties of the human cornea surpass those of the Viscoll collagen membrane by about six-fold. Still, the mechanical properties of the Viscoll collagen are sufficient for its manipulation and fixation with a surgical suture. This is discussed in more detail below. Table 1 summarizes the optical and mechanical properties of collagen membranes prepared from 3% (*w*/*v*) Viscoll collagen solution and stroma of the human cornea.

### 3.2. Suturability of the Membranes

To assess the suitability of the membranes for surgical practice, it was necessary to test them under ex vivo conditions during keratoplasty of the cadaveric eye. It was demonstrated that the Viscoll collagen membrane can be fixed with both interrupted and continuous sutures without macroscopic damage to the material (Appendix A). Suture and fixation of the material are possible using standard surgical instruments but require more careful handling; it is not recommended to exert strong pressure on the material when grasping with tweezers, as this can lead to damage.

### 3.3. Toxicology

Viscoll collagen membranes successfully passed all the necessary toxicological tests. When L929 mouse fibroblasts were cultured with extracts from Viscoll membrane samples, no cytotoxic effects were detected. This sensitizing effect was also studied in sexually mature albino guinea pigs using a method to maximize the sensitizing effect. None of the animals exhibited hypersensitive reactions. When the solution was instilled into the conjunctival sac of rabbits, no irritating effect on the cornea, iris, or conjunctiva was detected. In addition, no chemoses or pathological secretions were observed. Under the conditions of implantation of the collagen membrane samples, no general toxic effects were detected and there were no observable effects on the tissues surrounding the implant compared to those in the controls.

Extracts prepared in a sterile solution of 0.9% sodium chloride for injection did not show pyrogenic reactions when administered intravenously to rabbits. The total temperature increase did not exceed the permissible value (∑∆t) of ≤1.2 °C.

### 3.4. Clinical Results

Clinical pictures were recorded daily during the first stage of the study. In the early postoperative period, slight corneal oedema was observed in all 10 cases, which was most pronounced in the area of the surgical suture. There were no signs of inflammation, the moisture in the anterior chamber was transparent, and the iris actively reacted to light; 2 weeks after the operation, oedema had largely regressed. The subsequent clinical state was stable; the cornea remained transparent and no signs of inflammation were observed (Figure 3a). However, by the end of the 6th month of observation, against the background of sagging suture material, four rabbits showed signs of corneal neovascularization in the form of superficial “tassels” of newly formed vessels, stretching from the limbus to the operation area and into the suture area. After the removal of the sutures, the vascular reaction stopped and the newly formed vessels quickly became empty (Figure 4). In the second group, the clinical picture was similar, with slight oedema in the postoperative period associated with surgical trauma and subsequent stabilization. In rabbits from the second group, after 6 months of observation, a vascular reaction to the suture material was also noted, but was less pronounced.

### 3.5. OCT and In Vivo Confocal Microscopy Results

According to the OCT data, the collagen membrane was in close contact with the corneal stroma throughout the observation period (Figure 3b). One month after the operation, the average thickness of the central region of the cornea in the first group was 365 ± 23 µm, and the boundaries of the implant were visible, making it possible to estimate its thickness, which was 190 ± 15 µm. At 3 and 6 months after surgery, the boundaries of the implant and stroma were indistinct, which indicates good integration of the collagen membrane with the surrounding tissue. Compared to the first month, the average thickness of the central region of the cornea after 3 and 6 months decreased by 316 ± 19 µm and 305 ± 20 µm, respectively. The mean central corneal thickness of the non-operated corneas of the contralateral eye was 230 ± 20 μm.

IVCM was performed to evaluate the anatomical layers of the cornea and the membrane at the cellular level [8]. The surface epithelium and endothelium remained unchanged throughout the experiment and were comparable to those of non-operated corneas (Figure 5). After 6 months, the presence of sub-basal nerves was noted under the epithelium and in the central region of the cornea, indicating regeneration of the sub-basal nerve plexus. Two rabbits were noted to have slightly more macrophages than the other rabbits, with a clear trend towards a decrease by 6 months. The presence of linear structures, which appear due to apoptosis of the stromal cells [9], was observed at 1 and 3 months after the operation; by 6 months, their number had significantly decreased.

### 3.6. Histological Results

At 6 months after the surgery, the cornea was preserved throughout and was of uniform thickness. The cornea’s substantia propria had no visible changes throughout its thickness, and the anterior epithelium and border (Bowman) membranes did not change. The posterior (Descemet’s) membrane, as well as the adjacent stromal layer (Dua’s layer) were preserved, as was the posterior epithelium (endothelium). The implant was well traced along its entire length; it was not defibrillated and was fully integrated into the corneal stroma. At the border of the cornea with the limbus, there was scant inflammatory infiltration (predominantly perivascular) and slight hyalinosis along this side of the implant. In addition, host cell migration within the material was observed in the samples (Figure 6), which is in agreement with our previous results [5]. The results of immunohistochemical staining of corneal samples 6 months after surgery revealed single stromal cells expressing α-SMA in both groups (Figure 7).

## 4. Discussion

Given that more than 90% of the human cornea is in the stroma, the solution of creating a fully functional and biocompatible stromal replacement is key to both artificial cornea technology and the ability to restore a healthy stroma in patients with corneal blindness [3]. Therefore, active practical research is being conducted in this direction, among which corneal stroma tissue engineering with cell-free collagen hydrogels [10,11] is a promising approach with potential clinical application. The pioneers in this field are the Griffith et al. group, who were the first to conduct clinical trials of collagen hydrogels, which were prepared from recombinant type III collagen chemically cross-linked with EDC/N-hydroxysuccinimide to impart mechanical strength to the material [12]. Although this work demonstrated the safety of this approach and even the restoration of innervation, in subsequent studies from the same group, the absence of host cell migration into such material was noted four years after surgery in humans [13]; therefore, complete regeneration of the cornea has not been achieved thus far. There may be several reasons for this, but it is most likely that the chemical cross-linking of the collagen material significantly impedes the migration of cells into the implant and, as a result, complete regeneration of the corneal stroma is unattainable.

Many years of experience with collagen experimentation have shown that native highly purified collagen has extremely low antigenic properties and low inflammatory potential [14], leading to the adoption of collagen biomaterials in clinical practice. Therefore, when developing new collagen biomaterials, it is necessary to be guided by the following principle: the fewer structural and property changes to collagen during the process, the more likely it is to obtain an ideal collagen biomaterial. This principle is especially important if the goal of this study is to achieve the clinical application of the biomaterial.

Our approach to the creation of an artificial cornea is to reject the use of any chemical cross-links, as they can impair the biocompatibility of the entire implant [15] and hinder the process of host cell migration into the implant [16], or, in some cases, completely block it [13]. Therefore, to enhance the biomechanical characteristics of the implant, a solution of concentrated medical-grade Viscoll collagen I was used. In contrast to our previous work [5], we used a higher concentration of collagen to manufacture the membrane. This made it possible to increase the mechanical properties of the membrane compared to our previous work [5], but this was not enough to achieve the mechanical properties of the human cornea. It should be noted that this is not a critical disadvantage since the properties of the Viscoll collagen membrane are sufficient for its fixation with a classic surgical suture. At the same time, we had previously shown that its mechanical properties after implantation into the stroma of the cornea will inevitably increase due to the natural process of adaptation of the collagen membrane with the surrounding tissues [5]. Moreover, cells can also enhance the mechanical strength of collagen hydrogels [17]. Thus, the success of the regeneration of the damaged cornea will depend on how effectively the corneal cells will interact with the collagen membrane in vivo.

An important result obtained in this study was that active cell migration was observed in all implanted collagen membranes after 6 months. This is consistent with our previous in vitro results, which demonstrated that collagen hydrogels prepared from a concentrated collagen solution maintained a high survival rate of encapsulated cells, maintained their morphology and functional activity for at least 28 days, and did not interfere with cellular movement [17]. Our cellular migration results were also consistent with earlier in vivo work by our group [5]. These results are in stark contrast with those from the cross-linked recombinant human collagen membrane, where host cell migration into the hydrogel was not observed up to 4 years post operation [13], as well as with the data obtained for cross-linked porcine collagen in a rabbit study, in which only sparse cell migration into the hydrogel was observed at 6 months post surgery [10]. This may be evidence that the chemical cross-linking of collagen hydrogels impairs cell permeability in vivo.

In the study of corneal tissue regeneration, it is extremely important to investigate the issue of innervation of the restored area because nerve regeneration after corneal injury plays a decisive role in restoring normal function.

A study of the micromorphology of the cornea using laser confocal microscopy revealed that the removal of a portion of the stroma during anterior lamellar keratoplasty leads to the initial loss of nerves; however, in all experimental animals, nerve regeneration was confirmed at 6 months after the operation. Similar results were obtained during the implantation of chemically cross-linked collagen membranes into the corneal stroma of animals [10,11,18] and humans [13]. It is worth noting here that the process of reinnervation of implanted donor corneas in humans is very slow, lasting from one to several years [6,19]. The results obtained in this study may indicate that the Viscoll collagen membrane supports, or at least does not interfere with, the process of nerve repair in the cornea after layered keratoplasty.

According to our OCT data, after implantation, there was no migration or displacement of the collagen membrane implant; it took its position and fixed itself within the surrounding tissues. This behavior was also confirmed by histological examination, which showed that the collagen fibers of the stroma were closely intertwined with those of the implant, indicating a good survival rate.

Corneal neovascularization in the peripheral implanted zone was noted in four rabbits in the first group and two rabbits in the second group. Neovascularization is associated with sagging of the suture material due to the gradual integration of collagen with the corneal stroma; this involves natural cross-linking and strengthening of the biomechanics, which significantly weakens the sutures and causes sagging [20]. In such cases, the timely removal of the sutures completely resolves the problem and, according to the results of our study, this should be done 4–5 months after the operation.

Immunohistochemical staining of cryosectioned cornea samples revealed the presence of single α-SMA-positive stromal cells in both groups, whereas none were found in the intact cornea samples. Notably, the presence of these cells did not affect the transparency of the corneas. The presence of α-SMA-positive cells in the corneal stroma was likely induced by the operation itself, during which a part of the stroma was removed. The results provide evidence that the Viscoll collagen membrane does not have an irritating effect on the corneal tissue, which is also consistent with the results of the toxicological tests performed.

Our work has several limitations. (1) The experiments were carried out on a healthy cornea, in which a section of healthy stroma was surgically removed and a collagen membrane was implanted in its place. There is no doubt that the processes of tissue regeneration in various pathologies of the cornea, especially in case of chemical burns, will proceed differently. However, our results strongly suggest that the Viscoll collagen membrane may also be effective in the treatment of pathological conditions of the cornea. It should be noted that in this case, it may be necessary to develop new approaches for the treatment of various corneal pathologies using the Viscoll collagen membrane. (2) Short-term observation: Ideally, for a clear understanding of the reparative processes occurring in the damaged cornea during implantation of the Viscoll collagen membrane, a four-year follow-up is required. At the same time, it is extremely difficult to monitor rabbits that have undergone a full course of anesthesia for more than 6 months. At the same time, the processes of regeneration of the cornea in rabbits proceed faster than in humans. Therefore, to a certain extent, the results could be extrapolated to the regeneration of the human cornea.

Considering the proven safety of the Viscoll collagen membrane and its possible potential for the treatment of various corneal pathologies, especially under the conditions of a global shortage of donor tissue, our results can serve as the basis for future clinical trials.

## 5. Conclusions

In this study, a Viscoll collagen membrane prepared from a concentrated collagen solution was presented. Viscoll collagen membrane manufacturing technology can be easily scaled up to 5000 units, and the proposed packaging format allows for the maintenance of its key properties for at least 12 months. In addition to its excellent optical and mechanical properties, the Viscoll collagen membrane promotes active cell migration and can be implanted into the corneal stroma using tools and techniques that mimic those used in human corneal keratoplasty. From the aggregate of the presented data, it can be concluded that the Viscoll collagen membrane has the potential to solve the global problem of limited donor tissue for the treatment of corneal blindness. In addition to this study, we have demonstrated good biochemical properties of the Viscoll collagen membrane and the ability of suture fixation; therefore, the next step in our work will be deep anterior lamellar keratoplasty with the Viscoll collagen membrane.

## Figures and Tables

**Figure 1 polymers-14-04017-f001:**
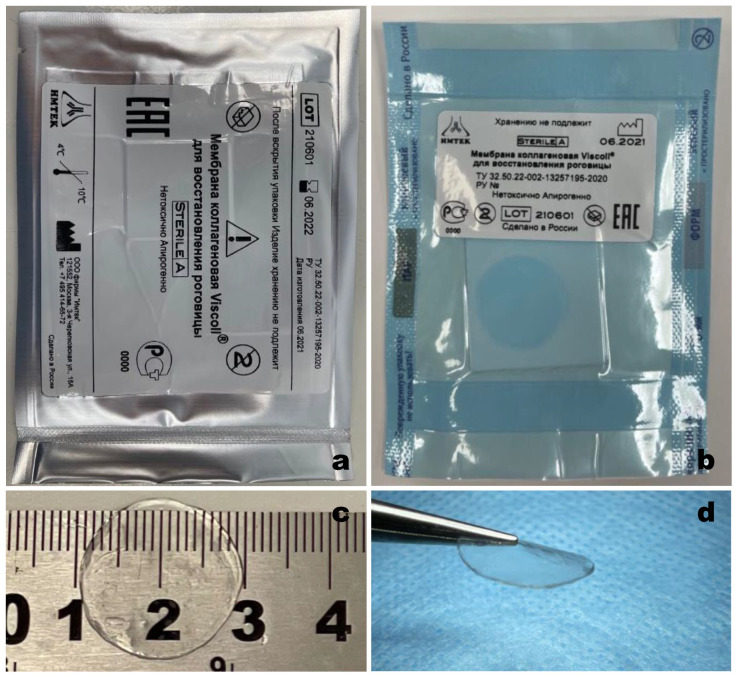
(**a**) Viscoll collagen membrane in vacuum packaging; (**b**) Viscoll collagen membrane in primary packaging; (**c**,**d**) views of the Viscoll collagen membrane ready for use.

**Figure 2 polymers-14-04017-f002:**
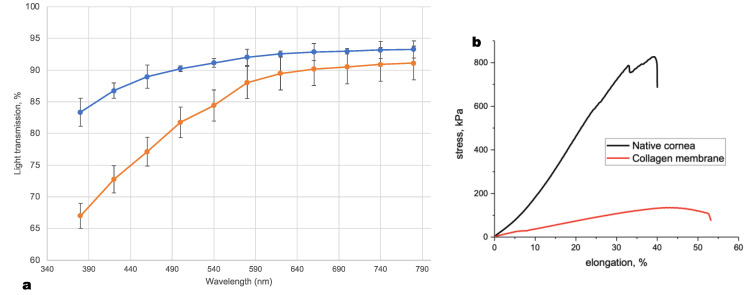
(**a**) Optical properties of the Viscoll collagen membrane (blue line) and stroma of human cornea (orange line). Samples thickness: 300 µm. (**b**) Characteristic stress–tension curves for Viscoll collagen membranes and human cornea samples.

**Figure 3 polymers-14-04017-f003:**
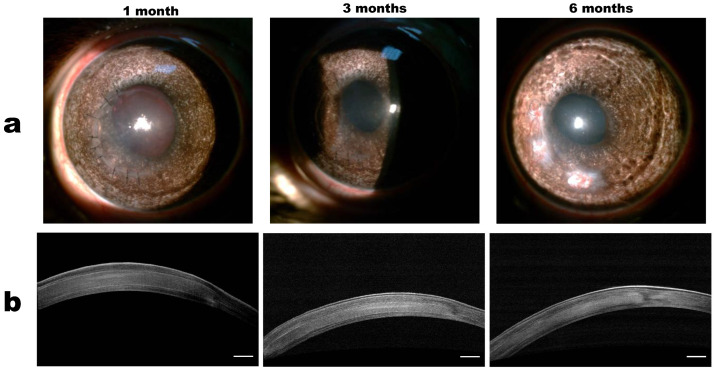
(**a**) Representative photographs of collagen membranes implanted in the stroma at different follow-up times. (**b**) Representative OCT images of rabbit corneas at different follow-up times. Scale bar = 250 μm.

**Figure 4 polymers-14-04017-f004:**
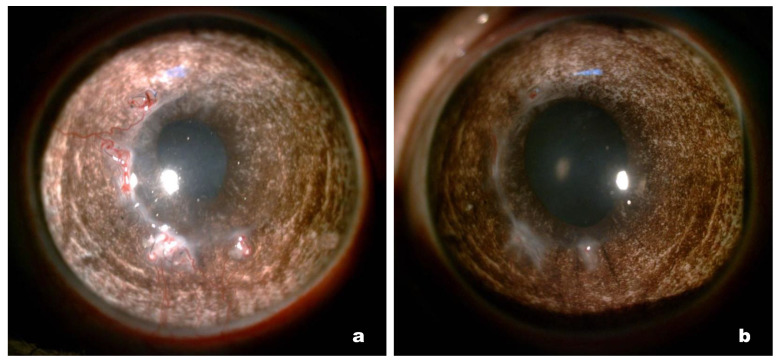
The role of sutures in corneal neovascularization. (**a**) Rabbit cornea with implanted membrane 6 months after surgery (before suture removal); (**b**) same cornea after removal of the sutures (1 week later).

**Figure 5 polymers-14-04017-f005:**
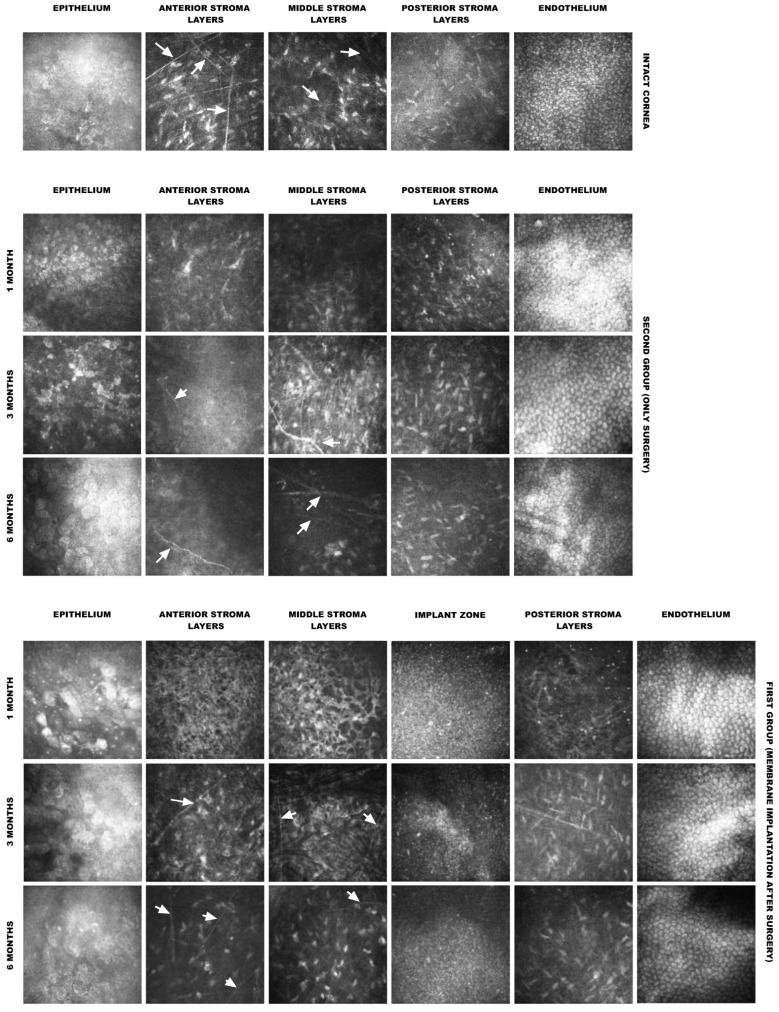
In vivo confocal microscope images of rabbit corneas implanted with the collagen membrane at different observation times. Arrows indicate nerves.

**Figure 6 polymers-14-04017-f006:**
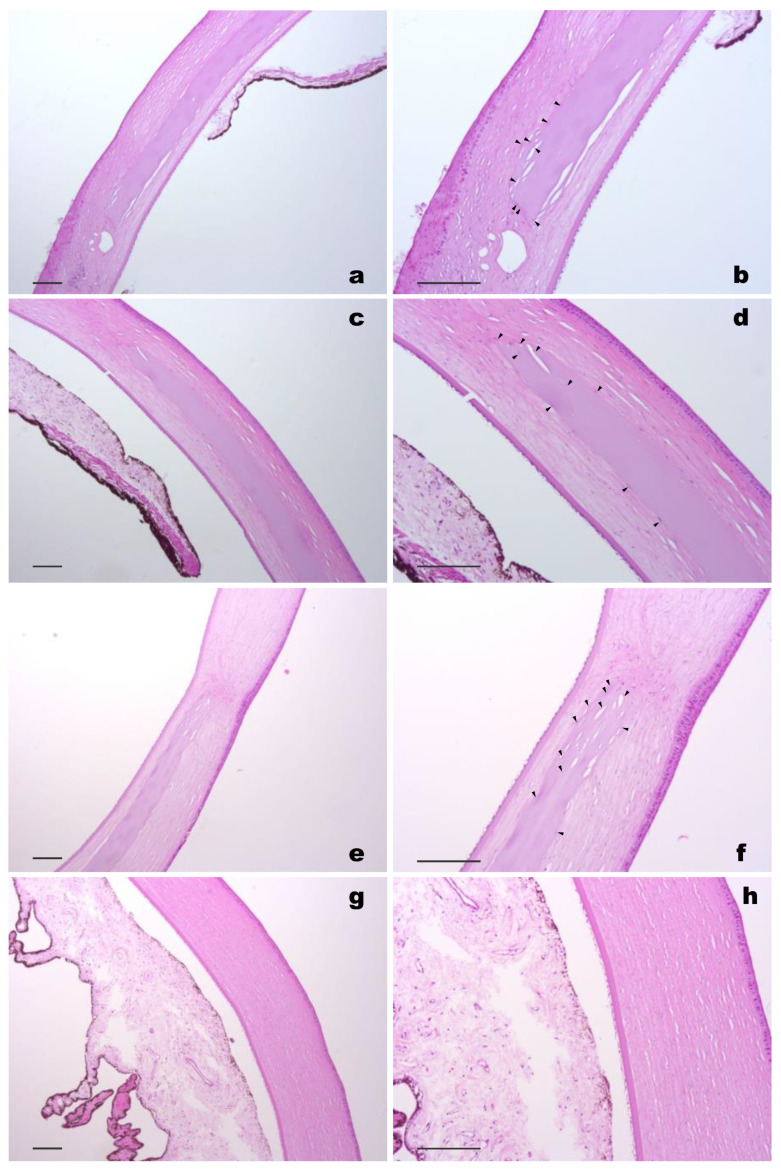
Hematoxylin and eosin staining of a cornea 180 days after the implantation of the collagen membrane: (**a**–**f**) first group, (**g**,**h**) intact cornea. Arrows indicate the migration of host cells into the membrane. Scale bar = 100 µm.

**Figure 7 polymers-14-04017-f007:**
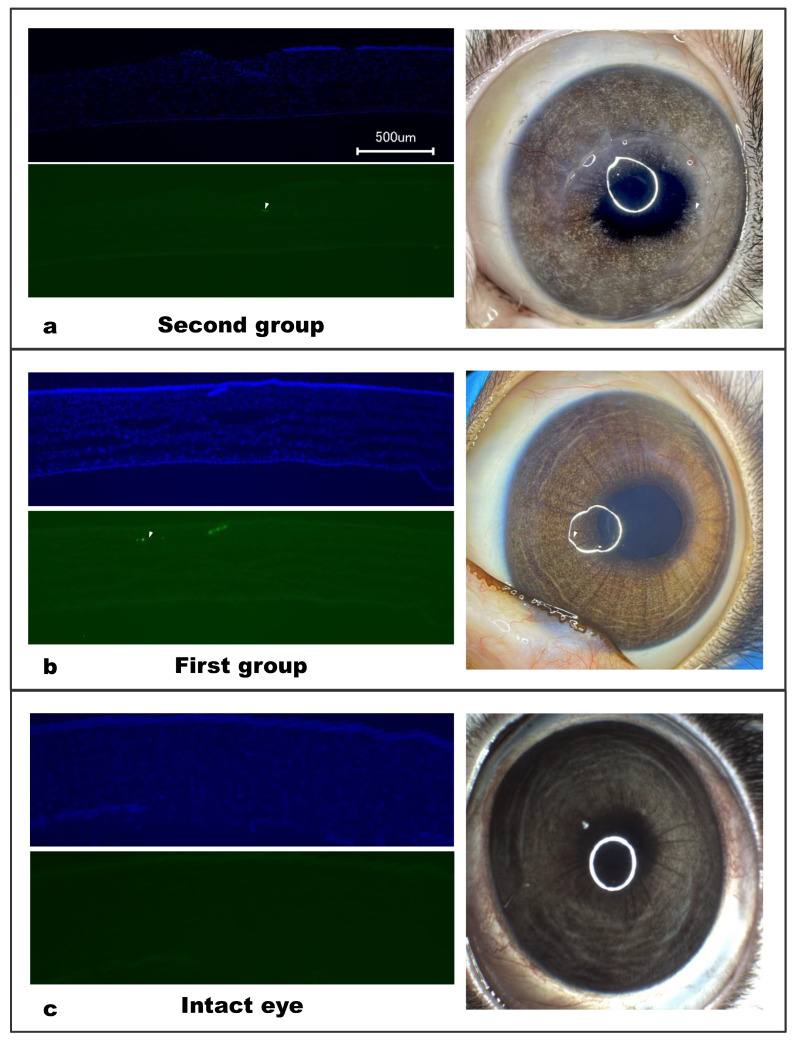
Representative photographs of rabbit eyes and their immunohistochemical analysis 6 months after surgery. (**a**) Second group (only surgery); (**b**) first group (surgery with collagen membrane implantation); (**c**) contralateral control eye. In both groups, the presence of α-SMA-positive fibroblasts was noted. Blue—DAPI staining (nuclei), green—α-SMA staining.

**Table 1 polymers-14-04017-t001:** Optical and mechanical properties of Viscoll collagen membrane made from 3% (*w*/*v*) collagen solution and stroma of the human cornea.

Sample	Thickness (mm)	Transparency (%)	Young Modulus (kPa)	Stress at Rupture (kPa)	Elongation at Rupture (%)
Viscoll Collagen membrane	0.3	83.3–93.3 (at λ from 380–780 nm)	467 ± 30	137 ± 11	41 ± 5
Stroma of human cornea	0.3	67.0–91.1 (at λ from 380–780 nm)	2859 ± 30	836 ± 6	41 ± 2

## Data Availability

The data presented in this study are available on request from the corresponding author.

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
