# Peer review of "Corneal Stroma Regeneration with Collagen-Based Hydrogel as an Artificial Stroma Equivalent: A Comprehensive In Vivo Study"

_polymers, 2022, doi:10.3390/polym14194017_

Round 1

Reviewer 1 Report

The current manuscript adds value to the previously reported results and envisaged to provide an improved product for corneal applications. I have some comments for the study as below:

1. The figures form and important part of the manuscript and I am very concerned with the quality of the figures, specifically Figure 5 with confocal images in grey scale.

2. The mechanical properties of the modified system are not conducted and provided.

3. Have the authors considered full corneal transplantation in the animal model?

Author Response

Comments to Author(s):

  • The figures form and important part of the manuscript and I am very concerned with the quality of the figures, specifically Figure 5 with confocal images in grey scale.

High quality pictures were sent in a separate zip file. Regardless of this, we improved the quality of all pictures in the text. Regarding Figure 5, it cannot be made in color, since the original raw images are black and white.

  • The mechanical properties of the modified system are not conducted and provided.

Thank you for your valuable comment. We have made significant changes in the experiment of evaluating the biomechanical characteristics of the Viscoll collagen membrane. We have added comparison of mechanical properties of  Viscoll collagen membrane with the natural human cornea. Detailed description of the methods was provided also. In addition, we have added a new Figure 2.B “Stress-Tension Characteristic Curves for Viscoll Collagen Membrane and Human Cornea Specimens” and updated data in table 1.

  • Have the authors considered full corneal transplantation in the animal model?

Yes, we have plans to perform full corneal transplantation in the animal model. However, we shall conduct this experiment after completed design of a functional artificial analogue of the cornea. That step involves combination of cell technologies with the offered Viscoll collagen membrane. This suggests formation of a coating layer of corneal endothelial cells on one side of the membrane. There should be a layer of either corneal limbal cells or corneal keratinocytes on the other side.

Reviewer 2 Report

General comments

The present manuscript is focused on the in vivo characterisation of a Viscoll collagen membrane for replacing damaged stromal tissue.

The Authors should better highlight the originality of the present paper, and improve the Materials and Methods section, as well as the Results and Discussion ones.

Moreover, an editing of English language and style is required. Some specific remarks and suggestions are reported below point by point.

1. Introduction

- The considerations “Donor cornea transplantation is the most common method of surgical treatment for most diseases of the cornea; however, the global shortage of donor tissue significantly complicates its use. Thus, the development of alternative approaches based on tissue engineering and regenerative medicine are necessary to solve this problem.” need to be corroborated with suitable references.

- The purpose of the work is clear, but the originality and added value to the scientific community of the present research paper has to be evidenced at the end of the Introduction section.

- It is strongly recommended to report a brief list of the performed characterisations.

2. Materials and Methods

2.1. Animal experiments

- The applied general anaesthesia (ketamine (50 mg/kg) and xylazine (15 mg/kg)) has to be specified also in this point.

2.2. Collagen membrane preparation

- For the sterile native porcine collagen solution, was the concentration in %w/v o % w/w?

- The used pressure up to vitrification has to be specified.

2.3. Optical properties of collagen membranes

- Usually the optical properties of solid samples, such as membranes and films, are measured in dry conditions.

- Moreover, the resolution has to be specified.

2.4. Toxicology

- Even if the tests standards are reported, the applied tests have to be briefly described.

- 2.6. Surgery

 For the reported percentage concentration, the kind has to be specified, if w/w or w/v or other.

2.8. Histological and immunohistological study

For the reported percentage concentration, the kind has to be specified, if w/w or w/v or other.

3. Results

3.1. Optical and physical properties

- For the reported percentage concentration, the kind has to be specified, if w/w or w/v or other.

- Even if the authors stated that “Figure 2 shows the transmittance as a function of wavelength in the 380–780 nm range for each membrane”, in Fig 2 only one curve is shown.

-The authors affirmed that “It was found that the mechanical properties of collagen membranes prepared from a

3% solution of Viscoll collagen exceeded the mechanical properties of collagen membranes”, but they should add some details and data, even if reported in the previous paper.

3.5. OCT and in vivo confocal microscopy results

- In Fig 5 the scale bar has to be added.

4. Discussion

-        The following consideration “There may be several reasons for this, but it is most likely that the chemical cross-linking of the collagen material significantly impedes the migration of cells into the implant and, as a result, complete regeneration of the corneal stroma is unattainable.” has to be supported with proper references.

-        Similarly, the statement “This is because collagen biomaterials are actively used in clinical practice. Therefore, when developing new collagen biomaterials, it is necessary to be guided by the following principle: the less structural and property changes to collagen during the process, the more likely it is to obtain an ideal collagen biomaterial.” This principle is especially important if the goal of this study is to achieve clinical application.” needs appropriate references.

-        Finally, the conclusion “Neovascularization is associated with sagging of the suture material due to the gradual integration of collagen with the corneal stroma; this involves natural cross-linking and strengthening of the biomechanics, which significantly weakens the sutures and causes sagging.“ has to be corroborated with suitable references.

Author Response

Reviewer 2:

Comments to Author(s):

  • The Authors should better highlight the originality of the present paper, and improve the Materials and Methods section, as well as the Results and Discussion ones. Moreover, an editing of English language and style is required. Some specific remarks and suggestions are reported below point by point.

Your other valued comments have been employed to make proper corrections as follows.

Introduction

  • Considerations “Donor cornea transplantation is the most common method of surgical treatment for most diseases of the cornea; however, the global shortage of donor tissue significantly complicates its use. Thus, the development of alternative approaches based on tissue engineering and regenerative medicine are necessary to solve this problem.” need to be corroborated with suitable references.

This consideration corroborated with the reference - Matthyssen, S.; Bogerd, B.W.den; Dhubgail, S. N.; Koppen, K.; Zakaria, N. Corneal regeneration: a review of stromal replacements. Acta Biomater 2018, 69, 31–41, doi:10.1016/j.actbio.2018.01.023

  • The purpose of the work is clear, but the originality and added value to the scientific community of the present research paper has to be evidenced at the end of the Introduction section. 

We have included additional text emphasizing the value and originality of the work in the introduction: “…..”

  • It is strongly recommended to report a brief list of the performed characterizations.

We have included a brief list of the performed characterizations in the Introduction: “…..”

Materials and Methods

  • The applied general anaesthesia (ketamine (50 mg/kg) and xylazine (15 mg/kg)) has to be specified also in this point.

We have included the referred data in this paragraph.

  • Collagen membrane preparation. For the sterile native porcine collagen solution, was the concentration in %w/v o % w/w? The used pressure up to vitrification has to be specified.

Description of the preparation collagen membranes has been expanded and improved. Indicating 3% w/v or 30 mg/ml precisely also.

  • Usually the optical properties of solid samples, such as membranes and films, are measured in dry conditions.

That is true for technological purpose. But it was important for us to imitate the cornea in its native hydrated state. So we choose to measure cornea optical properties and the optical properties of material that is supposed to be implanted in the cornea in a hydrated state.

Taking into account your remark we evaluated the optical properties of the Viscoll collagen membrane in hydrated state in the air, not submerged in water as before. The reference in this experiment was measurement of native stroma cut from a human donor cornea, 300 µm thick.

  • Moreover, the resolution has to be specified. 

The Fig 2 resolution has been increased.

  • Even if the tests standards are reported, the applied tests have to be briefly described.

We have included these data in the paragraph.

  • For the reported percentage concentration, the kind has to be specified, if w/w or w/v or other.

We have included this data in this paragraph

  • Histological and immunohistological study. For the reported percentage concentration, the kind has to be specified, if w/w or w/v or other.

Corrected

Results

  • Optical and physical properties. For the reported percentage concentration, the kind has to be specified, if w/w or w/v or other. Even if the authors stated that “Figure 2 shows the transmittance as a function of wavelength in the 380–780 nm range for each membrane”, in Fig 2 only one curve is shown. The authors affirmed that “It was found that the mechanical properties of collagen membranes prepared from a 3% solution of Viscoll collagen exceeded the mechanical properties of collagen membranes”, but they should add some details and data, even if reported in the previous paper. 

We have included required information and also we enforced our experiment by comparing the optical properties of 5 Viscoll collagen membrane with 5 normal human cornea stroma samples.

  • OCT and in vivo confocal microscopy results. In Fig 5 the scale bar has to be added.

Unfortunately, we cannot add a scale bar to Fig 5. These data were obtained using standard medical device for diagnostics. This device allows you to view the cornea in layer by layer. therefore, the point interest in the data obtained lies in the depth (Z axis) at which a particular image was taken. There is no option to measure linear dimensions (X,Y axis) in the software of this device, that is why there is no scale bar in this figure.

Discussion

  • The following consideration “There may be several reasons for this, but it is most likely that the chemical cross-linking of the collagen material significantly impedes the migration of cells into the implant and, as a result, complete regeneration of the corneal stroma is unattainable.” has to be supported with proper references.

Thank you for your comment, however, in our opinion, this statement does not require a specific reference. This consideration refers to the discussion of the clinical results presented in [13]: “Although this work demonstrated the safety of this approach and even the restoration of innervation, in subsequent studies from the same group, the absence of host cell migration into such material was noted four years after surgery in humans [13]”, which precedes this consideration in the text. Moreover, this statement is further supported by references in the next paragraph, which discusses the possible negative effects of chemical cross-linking - "Our approach to the creation of an artificial cornea is to reject the use of any chemical cross-links, as they can impair the biocompatibility of the entire implant [15] and hinder the process of host cell migration into the implant [16], or, in some cases, completely block it [13].”

  • Similarly, the statement “This is because collagen biomaterials are actively used in clinical practice. Therefore, when developing new collagen biomaterials, it is necessary to be guided by the following principle: the less structural and property changes to collagen during the process, the more likely it is to obtain an ideal collagen biomaterial.” This principle is especially important if the goal of this study is to achieve clinical application.” needs appropriate references.

Thank you for your comment, however, in our opinion, again, the statement does not need to be referred. It is the declaration of our view on working with collagen. At the same time, it should be noted that in the modern scientific literature there is no consensus on whether it is “good or bad” to use chemical crosslinks to stabilize collagen biomaterials, while a large number of mutually opposed data have been published. If we take into account the fact that “highly purified collagen has extremely low antigenic properties and low inflammatory potential [14].”, then our following statement logically follows from this statement - Therefore, when developing new collagen biomaterials, it is necessary to be guided by the following principle: the less structural and property changes to collagen during the process, the more likely it is to obtain an ideal collagen biomaterial.”. And if we take into account the results of work [13], in which the migration of host cells into the chemically cross-linked collagen membrane four years after implantation was completely absent, then this is an indirect proof of the veracity of our statement.

Anyhow we have also rephrased the first sentence to make it more clearly “Many years of experience with collagen experimentation have shown that native highly purified collagen has extremely low antigenic properties and low inflammatory potential [14], leading to the adoption of collagen biomaterials in clinical practice.”

  • Finally, the conclusion “Neovascularization is associated with sagging of the suture material due to the gradual integration of collagen with the corneal stroma; this involves natural cross-linking and strengthening of the biomechanics, which significantly weakens the sutures and causes sagging.“ has to be corroborated with suitable references.

Thank you for your comment. This consideration corroborated with the reference – Hsu, C.-C.; Chang, H.-M.; Lin, T.

C.; Hung, K.-H.; Chien, K.-H.; Chen, S.-Y.; Chen, S.-N.; Chen, Y.-T. Corneal Neovascularization and Contemporary

Antiangiogenic Therapeutics. J Chin Med Assoc 2015, 78, 323–330, doi:10.1016/j.jcma.2014.10.002.

Reviewer 3 Report

This paper offers an interesting topic for researchers in the area of corneal stroma tissue engineering. Moreover, it is a welcome addition to the current research which attempts to propose new collagen-based hydrogel formulations. As the challenge is to associate the bioactivity, optical and mechanical properties, this is a simple yet efficient paper regarding stromal replacement. However, there are 3 comments that still need to be addressed by the authors, and it is recommended to publish this paper after the modifications as below:

General comments

1.       The title should include the terms "hydrogel" or "collagen-based hydrogel".

2.       The authors should indicate the used device for vitrification (i.e., ...plastically compressed under constant pressure…).

3.       The authors should highlight the limitations of the study in the "conclusions" section.

Author Response

Reviewer 3:

Comments to Author(s):

  • The title should include the terms "hydrogel" or "collagen-based hydrogel". The title should include the terms "hydrogel" or "collagen-based hydrogel".

We agree with this comment. The title will be changed to: Corneal stroma regeneration with collagen-based hydrogel as an artificial stroma equivalent: a comprehensive in vivo study

  • The authors should indicate the used device for vitrification (i.e., ...plastically compressed under constant pressure…).

Description of the preparation of collagen membranes has been expanded and improved respectively.

  • The authors should highlight the limitations of the study in the "conclusions" section.

Our comment on limitations of the study has been added to the “conclusions” section.

Round 2

Reviewer 1 Report

The comments and concerns raised my the reviewer have been addressed.

Author Response

Spell check completed. Please, see revised manuscript

Reviewer 2 Report

General comments

The revised version of the manuscript looks very improved. Some minor revisions have to be applied, as reported below.

1. Introduction

- The conclusion “Results of these performed studies highlight a great potential of a Viscoll collagen membrane for its implementation in clinical practice for re-generation of the damaged cornea” does nto sound appropriate for the Introduction section, but more for the Abstract or Conclusions one.

2. Materials and Methods

2.4. Optical properties of collagen membranes and Human Cornea

- The request was relarted to the instrument resolution and not to the figure one.

-         

Author Response

  1. Introduction

- The conclusion “Results of these performed studies highlight a great potential of a Viscoll collagen membrane for its implementation in clinical practice for re-generation of the damaged cornea” does nto sound appropriate for the Introduction section, but more for the Abstract or Conclusions one.

We have removed this sentence from the text

  1. Materials and Methods

2.4. Optical properties of collagen membranes and Human Cornea

- The request was relarted to the instrument resolution and not to the figure one.

Below we provide some technical specifications of the device you are interested in:

  • Spectral slit width: 4 nm.
  • Wavelength setting error: no more than ±1 nm.
  • Reproducibility of the wavelength setting: ± 0.5 nm.
  • Limits of permissible absolute error when measuring spectral coefficients of directional transmittance, not more than: ±0.5%T (315-1000nm) and ±1.0%T (190-315nm).

All this information could be found on the official website of the manufacturer (https://ecohim.ru/en/good/spektrofotometry-i-aksessuary/spektrofotometr-pe-5400uf-s-derzhatelem-6-ti-kyuvet)